# Enhancing microalgal productivity through bioactive substances, light, and $CO_2$

**Andrea Llanes**[ID], **Wiston Quiñones**[ID], **Natalia Herrera**[ID]*

Group of Organic Chemistry of Natural Products, Institute of Chemistry, University of Antioquia, Medellín, Colombia

* natalia.herrera@udea.edu.co

## Abstract

Microalgae play a crucial role in ecosystems, from oxygen production to sustaining food webs and offering valuable applications in industry and environmental management. Increasing their productivity in terms of biomass yield, cultivation time, and nutritional or metabolite quality remains a major challenge. This study assessed the effects of 10 bioactive substances (including lactones, phytohormones, and natural extracts), four light wavelengths, and four $CO_2$ injection regimes on *Arthrospira platensis*, *Chlorella vulgaris*, *Ankistrodesmus falcatus*, and *Tetradesmus dimorphus*, using linear modeling and LSD (Least Significant Difference) post hoc tests. N-butyryl-DL-homoserine lactone significantly enhanced *A. platensis* growth (94.4%), while naphthaleneacetic acid and indole-3-butyric acid promoted *T. dimorphus* growth (138.1% and 115.5%, respectively). More accessible alternatives, such as *Aloe vera* and coconut water, also stimulated growth: *A. platensis*, *C. vulgaris*, and *A. falcatus* increased by 85.3%, 69.2%, and 87.7% with *Aloe vera*, while *T. dimorphus* increased 80.5% with coconut water. Regarding light quality, red light (600–700 nm) benefited *A. platensis* and *A. falcatus* (49.2% and 20.8%, respectively), whereas blue light (400–490 nm) favored *C. vulgaris* and *T. dimorphus* (57.7% and 31.5%, respectively). $CO_2$ injection further enhanced biomass production and carbon fixation, particularly in *C. vulgaris* and *A. falcatus* (73.5% and 53.5%, respectively). However, combined treatments did not produce additive effects, suggesting complex interactions. Overall, these findings demonstrate the potential of bioactive substances and environmental conditions to improve microalgal performance and highlight the importance of investigating synergistic effects and scalability for large-scale production.

## 1. Introduction

Microalgae can synthesize value-added products such as polysaccharides, proteins, lipids, and biologically active compounds that have important applications in the pharmaceutical, cosmetic, food, and biofuel production industries [1]. In addition, they play a crucial role in maintaining ecosystems, mitigating metal toxicity, preserving

**Data availability statement:** All relevant data are within the manuscript and its Supporting Information files.

**Funding:** The author(s) received no specific funding for this work.

**Competing interests:** The authors have declared that no competing interests exist.

aquatic and terrestrial biodiversity, treating wastewater, and, in a general sense, capturing $CO_2$ [2]. Microalgae biomass is a renewable and efficient resource that can be key to meeting the growing demand for sustainable alternatives, especially in sectors such as bioenergy, food security, and reducing the environmental footprint [3]. Due to this significant potential to play a significant role in various areas and industries, there has been considerable interest in optimizing microalgal biomass production.

Optimizing microalgal biomass production is crucial for its scalable and economically viable use [4]. One way to do this is by controlling some of the growth factors such as pH, temperature, salinity, nutrients, light intensity and quality, and the addition of inducing substances, among others [5,6]. Chunzhuk et al. [7] reported that increasing light intensity along with the addition of $CO_2$ can improve the production of *Arthrospira platensis, Chlorella ellipsoidea, Chlorella vulgaris, Gloeotila pulchra,* and *Elliptochloris subsphaerica.* Meanwhile, Xie et al. [8] y Seemashree et al. [9] reported that the addition of inducing substances such as phytohormones in the culture medium can also improve the production of *Chlorella vulgaris* (cv-31), *Porphyridium purpureum*, and *Dunaliella salina*.

Despite these advances, most studies have evaluated growth factors individually, with limited attention to their combined effects or to the use of alternative, low-cost inducers. Understanding how different factors interact is crucial, as responses are often species-specific and non-linear, and outcomes cannot be predicted from single-factor experiments alone. In particular, exploring the potential of natural extracts as sustainable bioactive substances, along with variations in light quality and $CO_2$ enrichment, could provide valuable insights for both fundamental physiology and applied large-scale cultivation [10,11]. Such approaches may help identify strategies that are not only effective in enhancing biomass productivity but also economically and environmentally feasible for industrial implementation [12].

Given the variability in species-specific responses, it is necessary to investigate how individual strains react to different factors and their interactions. Accordingly, this study aimed to assess the effects of 10 bioactive substances, four light wavelengths, and four $CO_2$ injection times—individually and in combination—on the growth and/or nutritional quality of *Chlorella vulgaris*, *Ankistrodesmus falcatus*, *Tetradesmus dimorphus*, and the cyanobacterium *Arthrospira platensis*, with the goal of improving biomass production.

## 2. Materials and methods

### 2.1. Obtaining and identifying microalgae

The cyanobacterium *Arthrospira platensis* was acquired from the Algae Culture Laboratory – Biology Department of the National University of Colombia. *Ankistrodesmus falcatus* was obtained from the Laboratory for Environmental Health Assessment and Promotion, Oswaldo Cruz Institute (Fiocruz), Brazil; *Chlorella vulgaris* and *Tetradesmus dimorphus* were isolated from different soils around the water reservoir in the village of La Palma in Carmen de Viboral, Antioquia, Colombia. To isolate the microalgae, 10 g of each soil sample containing *Chlorella* and *Tetradesmus* were first

weighed and transferred to 400 mL of BBM culture medium. The samples were kept on a shelf at room temperature for 3 days for conditioning.

Then, 50 mL of each culture was taken and centrifuged (Sigma 2-16P Universal Centrifuge, Sigma Laborzentrifugen GmbH, Osterode am Harz, Germany) at 5000 rpm (251 x g) for 5 minutes to concentrate the microalgae and remove unwanted particles. The precipitate obtained was resuspended in 400 mL of culture medium specific to each microalga and left to grow for 7 days. To clean the cultures of possible contaminants, three transfers were performed every 7 days, in which 10 mL of the previous culture was taken and finally transferred to 400 mL of new culture medium.

## 2.2. Microalgae cultivation and maintenance

Once cleaned and fully adapted, the cultures were maintained as follows: 500 mL Erlenmeyer flasks were used, each containing 400 mL of culture medium. The flasks were fitted with a rubber stopper containing two ports: one for connecting the aeration tube and another to release internal pressure. Aeration was continuous and maintained at 1.0 L/min. The culture media used were Zarrouk medium for *A. platensis*, BBM medium for *C. vulgaris* and *T. dimorphus*, and ASM-1 medium for *A. falcatus*.

The culture conditions were as follows: a 12 h:12 h light/dark photoperiod, temperature of 25 (±1) °C, light intensity of 100 µmol m$^{-2}$ s$^{-1}$, constant aeration at 1.0 L/min, pH 7 (±0.5) for BBM and ASM-1 media, and pH 9 (±0.5) for Zarrouk medium. Additionally, the culture media were renewed every 15 days.

For taxonomic identification, the morphological characteristics of each microalga were observed under a microscope (Nikon Eclipse E-200, Nikon Corporation, Tokyo, Japan), considering their structural features (size, organization, shape, pigments, motility) [13–15] following the Algaebase [16].

## 2.3. Exposure of microalgae to ten bioactive substances

Each microalga was exposed for 12 days to ten different bioactive substances, as detailed in Table 1 and illustrated in Fig 1. The assays were conducted under the culture conditions described in section 2.2.

## 2.4. Exposure to different wavelengths

Each microalga was exposed to four different wavelengths for 12 days. The assays were conducted under the culture conditions described in section 2.2, except with a continuous photoperiod of 24:0 h (light/dark) to maximize growth rate and biomass productivity, and to isolate the effect of light spectrum as the experimental variable while avoiding the influence of light–dark cycles [17–19]. To achieve this, the inner walls of the cultivation chambers were lined with reflective mirror film, and a commercial RGB LED strip with Bluetooth control was used to adjust each target wavelength. Based on manufacturer specifications (≈300–500 lm m$^{-1}$, 4–6 m total strip length per chamber) and enclosure geometry, the photon flux density was estimated to be in the range of 80–150 µmol m$^{-2}$ s$^{-1}$. The wavelengths tested were: blue light (400–490 nm, L1), red light (600–700 nm, L2), green light (490–550 nm, L3), and yellow light (570–580 nm, L4). Additionally, a white light control (400–700 nm) was included. All experiments were conducted in triplicate.

## 2.5. Exposure of microalgae to four $CO_2$ injection times

Each microalga was exposed to $CO_2$ at a flow rate of 1.0 L/min (99.95%, Messer SE & Co. KGaA) at times T1 = 30 s, T2 = 60 s, T3 = 90 s, and T4 = 120 s, for 12 days, every 24 hours, under the culture conditions described in section 2.2. Additionally, a control without the addition of $CO_2$ was used; each test was performed in triplicate. The final biomass of each culture was harvested by sedimentation for *C. vulgaris*, *T. dimorphus,* and *A. falcatus*, and using a 15 µm filter mesh for *A. platensis*. The biomass was washed with Milli-Q water, freeze-dried (LABCONCO FreeZone 12L, Labconco Corporation, Kansas City, MO, USA), and the dry weight was determined on a balance.

**Table 1. Bioactive substances used as potential inducers of microalgae growth evaluated.**

| Type of bioactive substance | Bioactive substance | Code | Origin |
|---|---|---|---|
| Lactone | N-Butiril-DL-homoserine lactone | SA | CAS: 98426-48-3, (Sigma-Aldrich, St. Louis, MO, USA) |
| Lactone | L-hydrochloride homoserine lactone | SB | CAS: 2185-02-6, (Sigma-Aldrich, St. Louis, MO, USA) |
| Phytohormone | Indol-3-butíric acid | SC | CAS: 133-32-4, (Spectrum Chemical, Gardena, CA, USA) |
| Phytohormone | 1-naphtalenacético acid | SD | CAS: 86-87-3, (Sigma-Aldrich, St. Louis, MO, USA) |
| Phytohormone | Indol-3-acético acid | SE | CAS: 87-51-4, (Millipore, Burlington, MA, USA) |
| Phytohormone | Salicylic acid | SF | CAS: 69-72-7, (Sigma-Aldrich, St. Louis, MO, USA) |
| Plant origin | Coconut water | SG | Local market: coconut water was obtained directly from the fruit and filtered with a 20 µm membrane to remove impurities (no further chemical treatment). Final test volumes were 400 mL; working concentrations were prepared v/v. |
| Plant origin | *Aloe vera (Aloe Barbadensis)* | SH | Local market: commercial *Aloe vera* crystals were homogenized in a food processor with distilled water at a ratio of 70% crystals: 30% distilled water to facilitate dissolution. The resulting crude *Aloe vera* preparation was used without further purification. Final test volumes were 400 mL; working concentrations were prepared v/v. |
| Plant origin | Lentil sprout extract (*Lens culinaris*) | SI | Local market: commercial lentils (Lens culinaris) were germinated until sprouting. A total of 100 g of germinated lentils were homogenized in a food processor and subjected to ethanolic extraction with 100 mL of 80% ethanol (v/v) under sonication for 20 min at 80 Hz. The mixture was filtered through a fine mesh to remove solids, and the filtrate was concentrated by rotary evaporation at 40 °C until complete ethanol removal, yielding a dry powder. This powder was subsequently used to prepare the working concentrations. |
| Plant origin | Sargassum extract (*Sargassum* spp.) | SJ | Obtained from the University of Sinú (Cartagena, Colombia). The sargassum was washed with distilled water to remove surface impurities and possible contaminants, then dried for 48 h under direct sunlight. The dried material was ground and subjected to ethanolic extraction using 100 g of dry biomass in 250 mL of 80% ethanol (v/v), kept under sonication for 20 min at 80 Hz. The resulting extract was concentrated by rotary evaporation at 40 °C until complete removal of the ethanol, yielding a dry powder. This powder was subsequently used to prepare the working concentrations. |

## 2.6. Evaluation of microalgae cell growth

The effect of each of the 10 bioactive substances, wavelengths, and $CO_2$ on microalgae growth was measured using four calibration curves made for each microalgae species. The curves were constructed as follows: Starting with a known number of cells at time zero, the progress of the culture was monitored with a Multiskan Spectrum (Thermo Scientific, Waltham, MA, USA) at 630 nm. During the first 15 days, measurements were taken every 24 hours, and from day 16 to day 30, they were taken every 48 hours. In addition, counts were performed in a Neubauer chamber (Marienfeld, Lauda-Königshofen, Germany) for *C. vulgaris*, *T. dimorphus*, and *A. falcatus*, and in a Sedgewick Rafter counting chamber (Pyser Optics, Edenbridge, UK) for *A. platensis*. Then, a calibration curve was constructed from which the equation relating the variables absorbance and cells/mL was obtained.

To evaluate the effect of the 10 bioactive substances, wavelengths, and $CO_2$ on microalgae growth, three samples were taken from each culture: one on day zero, as initial data, and two at the end of each experimental test, after 12 days, as final data. Of the two final samples, one was used to quantify cell growth by spectrophotometry, applying the equation obtained from each calibration curve. For this purpose, 300 µL of each culture was transferred to a 96-well plate, and the absorbance was measured using a wavelength reader [20]. Each measurement was performed in triplicate. The second sample was used for microscoPGI observations with a microscope (Nikon, Eclipse E-200, Tokyo, Japan) to evaluate possible changes in cell morphology (S1 File).

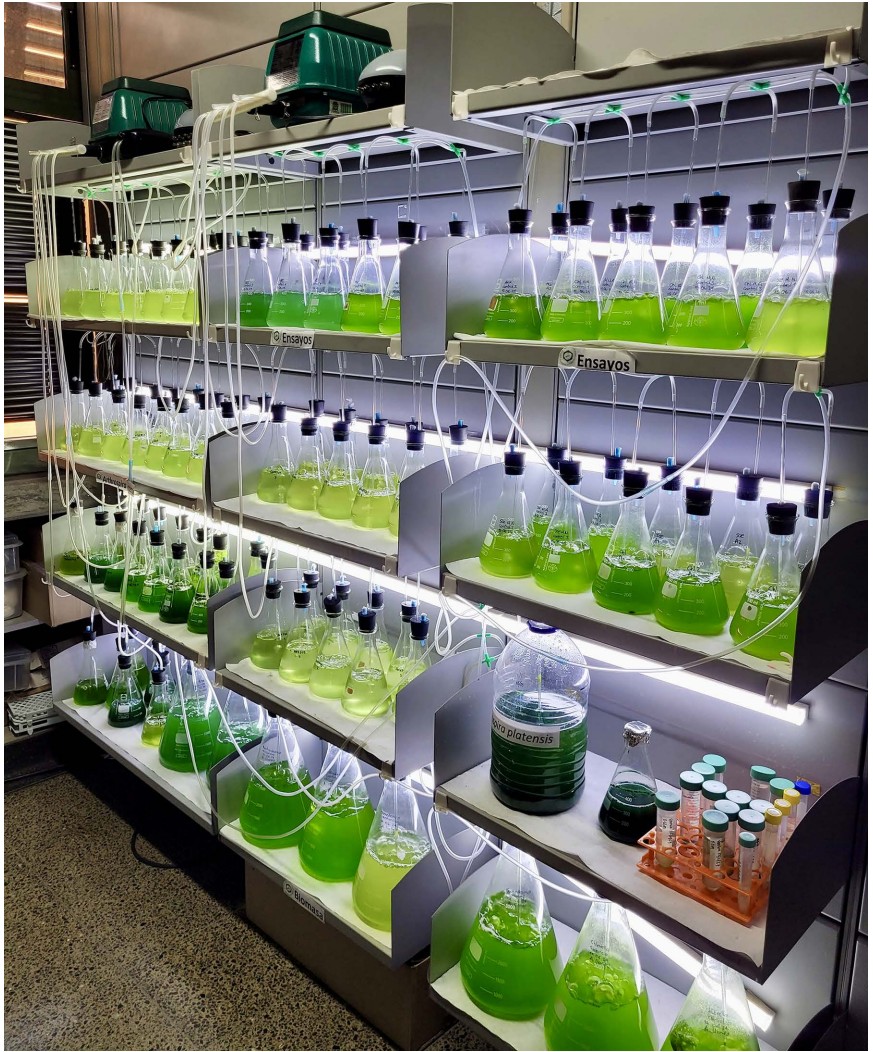

**Fig 1. Experimental setup of the bioactive substance exposure assays.** Lactones, phytohormones, lentil sprout extract, and sargassum extract were used at final concentrations: C1 = 0.1 μg/mL, C2 = 1.0 μg/mL, C3 = 5.0 μg/mL, and C4 = 10.0 μg/mL. Four concentrations were also tested for coconut water and Aloe vera: C1 = 1%, C2 = 3%, C3 = 7%, and C4 = 10% v/v. All treatments were applied in a final culture volume of 400 mL (in 500 mL Erlenmeyer flasks). A negative control containing only culture medium was included. Each assay was performed in triplicate.

In addition, the percentage of growth induction (PGI) of each microalga exposed to each bioactive substance, wavelength, and $CO_2$ was calculated using the following equation [20]:

$$PIC = \frac{\ln T_{final} - \ln control_{final}}{\ln control_{final}} * 100$$

Where T = Treatment, corresponds to the densities of cells exposed and not exposed (control) to the different bioactive substances, and ln = Natural logarithm.

Finally, $CO_2$ fixation (Fi) was calculated from the dry weight of the microalgae biomass and the carbon content of each species (% C), determined in the total oxidizable organic carbon analyses, using the following equation (adapted from Wu et al. [21]):

$$F_i = P_i * \%C * \frac{44}{12}$$

Where Fi is the $CO_2$ fixation for day i and Pi is the productivity obtained for day i, in discontinuous cultures.

## 2.7. Evaluation of the effect of various factors (bioactive substances, wavelength, and $CO_2$) and their combinations

Based on the results obtained in sections 2.3, 2.4 and 2.5 (bioactive substances, light wavelengths, and $CO_2$, respectively), the effects of these factors—considered inducers of growth and biomass production in microalgae according to their individual outcomes—were evaluated both individually and in combination. To this end, the four microalgal species were exposed to seven culture conditions for 12 days under the general conditions described in section 2.2. Growth was measured spectrophotometrically, as previously described (section 2.6), using in each case the optimal parameters determined in the preceding experiments. The seven conditions evaluated were:

• Control: Standard culture conditions (section 2.2)

• Condition 1 (C1): Bioactive substance

• Condition 2 (C2): Light wavelength

• Condition 3 (C3): $CO_2$ exposure time

• Condition 4 (C4): Bioactive substance, light wavelength, and $CO_2$ exposure time

• Condition 5 (C5): Bioactive substance and light wavelength

• Condition 6 (C6): Bioactive substance and $CO_2$ exposure time

• Condition 7 (C7): Light wavelength and $CO_2$ exposure time

## 2.8. Analysis of the nutritional composition of the biomass of each microalga

For the optimal treatments identified in the results of section 2.7, scaling was performed in 3 L Erlenmeyer flasks of 2.5 L, using 400 mL of culture from a 12-day stock of each microalga. The control consisted of scaling with constant aeration at 1 L/min, maintaining the culture conditions described in section 2.2. After 30 days of culture, the biomass produced was collected. The final biomass of each culture was harvested by sedimentation for *C. vulgaris, T. dimorphus,* and *A. falcatus*, and using a 15 µm filter mesh for *A. platensis*. The biomass was washed with Milli-Q water, freeze-dried (LAB-CONCO FreeZone 12L, Labconco Corporation, Kansas City, MO, USA) (pre-frozen at −20 °C for 24 h; operating vacuum 0.2–0.4 mbar), and the dry weight was determined on a balance.

The lyophilized biomass of each cultivated microalga was analyzed for carbon, nitrogen, and mineral content. Total oxidizable organic carbon was determined using the Oxidant-Titrimetric method (NTC 5167); nitrogen was analyzed by the volumetric Kjeldahl method (ISO 5983); phosphorus was quantified using the UV–VIS spectrophotometric method (NTC 4981); and minerals including calcium, copper, iron, magnesium, manganese, potassium, sodium, and zinc were analyzed using the atomic absorption spectrophotometric method (NTC 5151). It is worth noting that these mineral elements are not synthesized by microalgae but originate from the culture medium; however, their accumulation in the biomass may reflect

species-specific differences in uptake and bioaccumulation capacity, which is of interest from both nutritional and biotechnological perspectives [22].

## 2.9. Statistical analysis

Initially, the data were normalized using the natural logarithm to use the same scale for all data. A linear model was created, and an ANOVA (Analysis of Variance) was applied with a significance level of 0.05. The assumptions of normality (Kolmogorov-Smirnov test), independence (Durbin Watson test), and homoscedasticity (Breusch Pagan test) were validated. After performing ANOVA tests, significant differences were found, so it was decided to use the post hoc LSD (Least Significant Difference) test for multiple comparison analysis. All tests were analyzed at a significant level of 0.05 and a confidence level of 95%. For all analyses, RStudio 9.2 statistical software was used, with R 4.3 processing, Core Team (2023). The data were analyzed using a linear model, which showed a good fit in all cases according to the $R^2$ values and met most of the assumptions (Table A in S3 File).

## 3. Results

### 3.1. Effect of the evaluated factors (bioactive substances, wavelength, and $CO_2$) and their combinations on the growth of *Arthrospira platensis*

Initially, it was found that 1.0 µg/mL of N-Butyryl-DL-homoserine lactone (SA) has a positive effect on the growth of *A. platensis*, with a PGI = 94.4% and statistically significant differences compared to the other bioactive substances (p = 3.87e-12, Fig 2A). This statistical significance (letter "a" according to the LSD post hoc test) is shared with other treatments, such as *Aloe vera* (SH) at 3% (p = 6.38e-12) with a PGI = 91.9% and 7% *Aloe vera* (SH) (p = 3.23e-11) with a PGI = 88.1%. In addition, exposure to red light (600−700 nm, L2) generated an appreciable increase in growth, with a PGI of 49.2%, with statistically significant differences compared to the other wavelengths (p = 0.000125; Fig 2B). Finally, $CO_2$ injection (60 s, T2) also promoted significant growth, reaching a PGI of 41.2% (p = 1.12e-07; Fig 2C). In this case, 0.230 g of $CO_2$ was fixed, representing an increase of 0.068 g compared to the control (0.162 g).

Based on the results, *Aloe vera* was chosen for further testing as an inducing substance because it is a more accessible alternative compared to N-Butyryl-DL-homoserine lactone. Although the latter showed the highest effect (PGI 94.4%), *Aloe vera* at 3% achieved a PGI of 91.9%, with no statistically significant differences between the two. In evaluating the effect of the factors and their combinations, 3% *Aloe vera* (Condition 1) was statistically significant compared to the other conditions (p = <2.14e-10, Fig 3). Furthermore, Table 2 shows that the nutritional profile of *A. platensis* presented slight variations under the effect of 3% *Aloe vera*, with increases in calcium, phosphorus, iron and zinc content.

### 3.2. Effect of materials and conditions (bioactive substances, wavelength, and $CO_2$) and their combinations on the growth of *Chlorella vulgaris*

Initially, it was found that 3% *Aloe vera* (SH) has a positive effect on the growth of *C. vulgaris*, with a PGI = 76.2% and statistically significant differences compared to the other bioactive substances (p < 2.0e-16, Fig 4A). This statistical significance is shared with other treatments such as L homoserine lactone (SB) at 10.0 µg/mL (p = <2.0e-16) with a PGI = 76.0% and coconut water (SG) at 1% (p = <2.0e-16) with a PGI = 73.6%. In addition, exposure to blue light (400−490 nm, L1) generated a notable increase in growth, with a PGI = 57.7% with statistically significant differences compared to the other wavelengths (p = 2.47e-08; Fig 4B). Finally, $CO_2$ injection (60 s, T2) also promoted significant growth, reaching a PGI of 73.5% (p = 1.67e-08; Fig 4C). In this case, 0.384 g of $CO_2$ was fixed, representing an increase of 0.166 g compared to the control (0.218 g).

In evaluating the effect of individual materials and conditions, as well as their combinations, 3% *Aloe vera* (Condition 1) was statistically significant compared to the other conditions (p = 2.46e-12, Fig 5). In addition, Table 2 shows that the

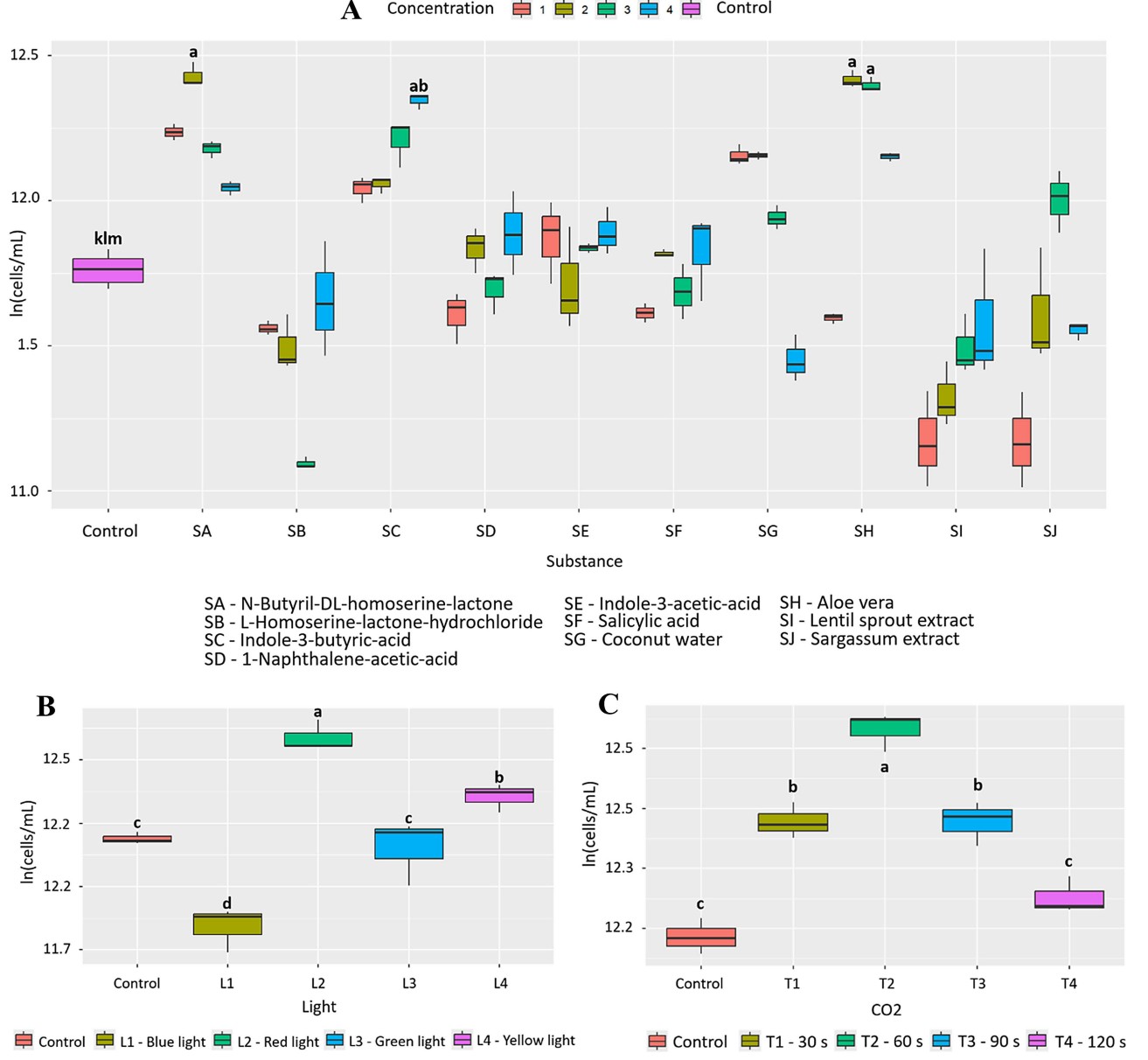

**Fig 2. Effect of the factors evaluated on the growth of *A. platensis*.** (A) 10 bioactive substances evaluated at 4 concentrations (B) 4 wavelengths and (C) 4 $CO_2$ injection times. Letters indicate statistically significant differences between treatments ($p < 0.05$).

nutritional profile of *C. vulgaris* exhibited a increase in total oxidizable organic carbon and potassium with 3% *Aloe vera*, accompanied by slight decreases in calcium, phosphorus, iron, and zinc content.

### 3.3. Effect of materials and conditions (bioactive substances, wavelength, and $CO_2$) and their combinations on the growth of *Ankistrodesmus falcatus*

Initially, it was found that 1% *Aloe vera* (SH) has a positive effect on the growth of *A. falcatus*, with a PGI = 97.4% and statistically significant differences compared to the other bioactive substances (p = 2.0e-16, Fig 6A). This statistical

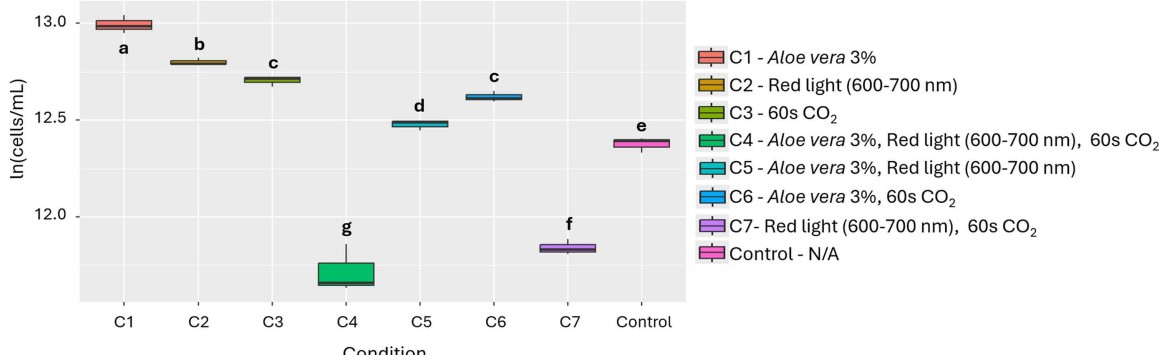

**Fig 3. Effect of the individual and combined use of the 3 evaluated factors on the growth of _A. platensis_.** Different letters indicate statistically significant differences among treatments (p<0.05).

**Table 2. Carbon, nitrogen, and mineral content of the best culture conditions for each microalga.**

| Analysis | _A. platensis_ | | _C. vulgaris_ | | _A. falcatus_ | | _T. dimorphus_ | |
|---|---|---|---|---|---|---|---|---|
| | Control | _Aloe vera_ 3% | Control | _Aloe vera_ 3% | Control | _Aloe vera_ 1% | Control | Coconut water 3% |
| Total oxidizable organic carbon | 34.3% | 34.3% | 34.3% | 38% | 45.0% | 46.3% | 34.0% | 33.1% |
| Calcium (mg/kg) | 19 | 190 | 190 | 181 | 40 | 39 | 197 | 345 |
| Cooper (mg/kg) | ND | <5 | <5 | <5 | ND | ND | <5 | <5 |
| Phosphorous (mg/kg) | <100 | 142 | 142 | 123 | <100 | <100 | 171 | 458 |
| Iron (mg/kg) | <5 | 8 | 8 | 5 | <5 | <5 | 8 | 9 |
| Magnesium (mg/kg) | <50 | <50 | <50 | <50 | <50 | <50 | 50 | 101 |
| Manganese (mg/kg) | <5 | <5 | <5 | <5 | <5 | <5 | <5 | <5 |
| Nitrogen (g/100g) | 10.9 | 9.2 | 6.9 | 6.1 | 5.7 | 5.9 | 5.5 | 5.3 |
| Potassium (g/100g) | 0.67 | 0.56 | 0.56 | 0.60 | 0.55 | 0.64 | 0.58 | 0.61 |
| Sodium (mg/kg) | <500 | <500 | <500 | <500 | <500 | <500 | <500 | <500 |
| Zinc (mg/kg) | ND | 15 | 15 | 11 | ND | ND | 15 | 21 |

ND: not detected

significance is shared with other treatments such as 1% coconut water (SG) (p=2.22e-13) with a PGI=76.6% and 3% _Aloe vera_ (SH) (p=5.93e-13) with a PGI=74.3%. In addition, exposure to red light (600−700 nm, L2) generated a slight increase in growth, with a PGI of 20.8%, with statistically significant differences compared to the other wavelengths (p=0.00177; Fig 6B). Finally, $CO_2$ injection (30s, T1) also promoted high growth, reaching a PGI of 53.5% (p=6.38e-08, Fig 6C). In this case, 0.271 g of $CO_2$ was fixed, representing an increase of 0.101 g compared to the control (0.154 g).

In evaluating the effect of materials and conditions, as well as their combinations, 1% _Aloe vera_ (Condition 1) was statistically significant compared to the other conditions (p=3.95e-15, Fig 7). Furthermore, Table 2 shows that the nutritional profile of _A. falcatus_ displayed minor increases in total oxidizable organic carbon and potassium content under the effect of 1% _Aloe vera_.

### 3.4. Effect of materials and conditions (bioactive substances, wavelength, and $CO_2$) and their combinations on the growth of _Tetradesmus dimorphus_

Initially, it was found that 10.0 µg/mL of 1-naphthaleneacetic acid (SD) has a positive effect on the growth of _T. dimorphus_, with a PGI=138.1% and statistically significant differences compared to the other bioactive substances (p=2.39e-16, Fig 8A).

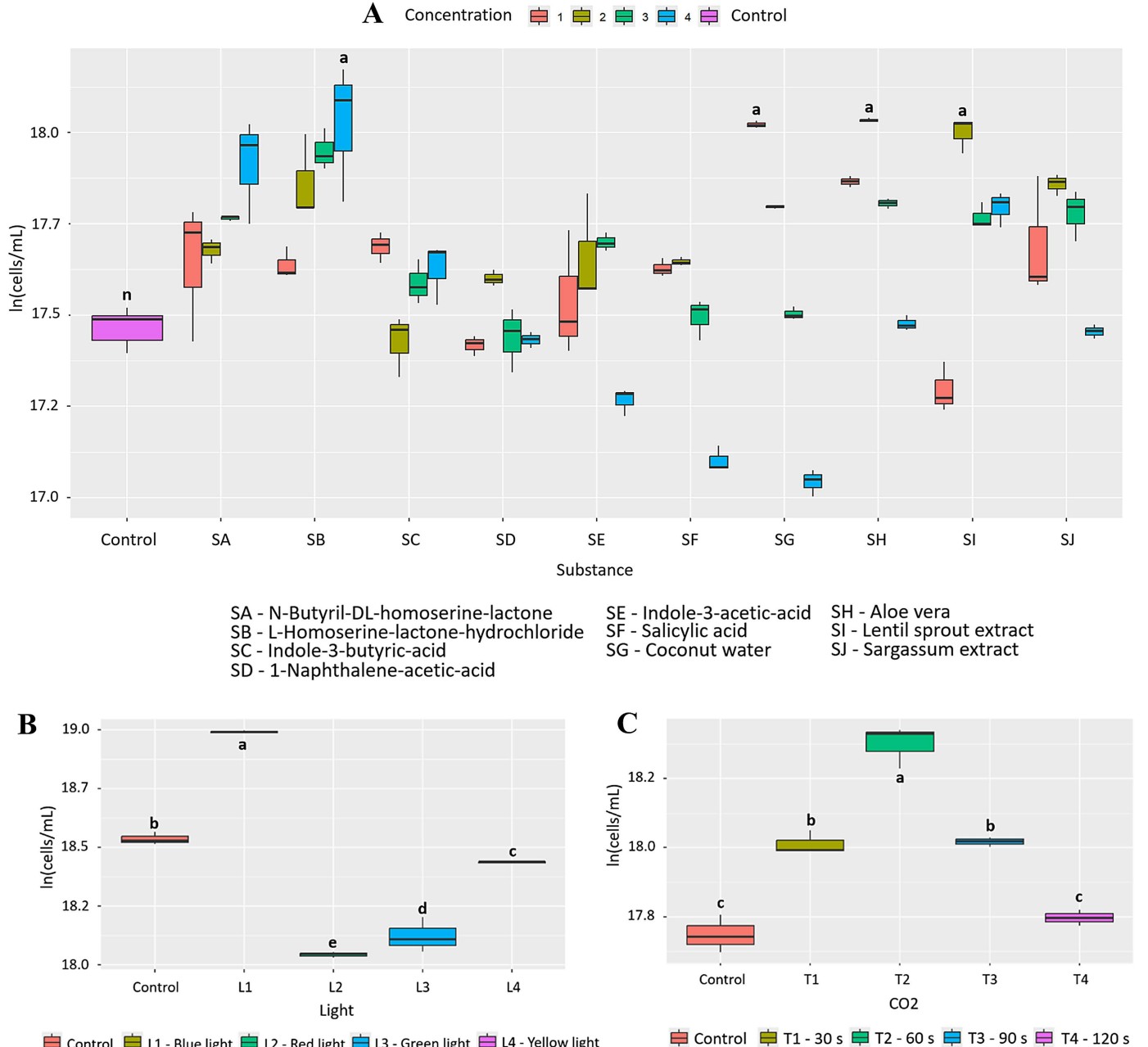

**Fig 4. Effect of the factors evaluated on the growth of *C. vulgaris*.** (A) 10 bioactive substances evaluated at 4 concentrations (B) 4 wavelengths and (C) 4 $CO_2$ injection times. Letters indicate statistically significant differences between treatments ($p < 0.05$).

This statistical significance is shared with other treatments such as indole-3-butyric acid (SC) 10.0 µg/mL ($p = 1.91e-13$) with a PGI = 115.5% and coconut water (SG) 3% ($p = 1.90e-11$) with a PGI = 93.8%. Additionally, exposure to blue light (400−490 nm, L2) generated an appreciable increase in growth, with a PGI of 31.5%, with statistically significant differences compared to the other wavelengths ($p = 1.01e-06$; Fig 8B). Finally, $CO_2$ injection (60 s, T2) slightly promoted growth, reaching a PGI of 21.7% ($p = 0.000824$, Fig 8C). In this case, 0.196 g of $CO_2$ was fixed, representing an increase of 0.043 g compared to the control (0.096 g).

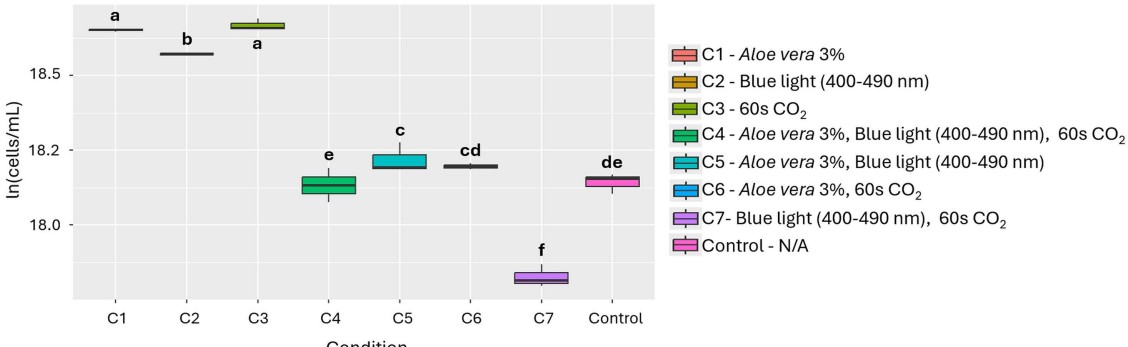

**Fig 5. Effect of the individual and combined use of the 3 evaluated factors on the growth of *C. vulgaris*.** Different letters indicate statistically significant differences among treatments (p < 0.05).

Based on the results and like aloe, coconut water (SG) (PGI = 93.8%) was chosen as an inducer because it is a more accessible alternative to other bioactive substances such as 1-naphthaleneacetic acid (SD, 138.1%) and indole-3-butyric acid (SC, 115.5%). Although its effect on growth is lower, it still shows a significant difference (p = 1.90e-11), which justifies its selection in this study. In the evaluation of the effect of the factors and their combinations, 3% coconut water (Condition 1) was statistically significant compared to the other conditions (p = 2.50e-10, Fig 9). In addition, Table 2 shows that the nutritional profile of *T. dimorphus* presented increases in calcium, phosphorus, iron, magnesium, potassium, and zinc content, accompanied by a slight decrease in oxidizable organic carbon content.

## 4. Discussion

This study evaluated the impact of different cultivation conditions on *Arthrospira platensis, Chlorella vulgaris, Ankistrodesmus falcatus,* and *Tetradesmus dimorphus* to optimize their growth and/or nutritional quality. Conditions such as the use of bioactive substances, wavelength, and $CO_2$ injection were analyzed, allowing the optimal conditions for each species to be identified.

The conditions evaluated in this study were selected based on their potential to induce microalgae growth and improve biomass quality. In particular, phytohormones and lactones were used, compounds widely recognized for their ability to regulate key physiological processes in plants and microalgae. Studies have shown that phytohormones such as indole-3-acetic acid, salicylic acid, indole-3-butyric acid, and naphthalene acetic acid, among others, can stimulate cell growth and the production of metabolites such as proteins, lipids, chlorophylls, and antioxidants [23–25]. Likewise, lactones (especially N-acylhomoserine lactones) have been studied as growth inducers in various microalgae species, probably due to their role in intercellular communication and the modulation of physiological responses [20,26,27]; which were shown to have an inducing effect on the growth of different microalgae studied.

In addition, natural extracts such as *Aloe vera* and coconut water were included due to their previous use, attributed to their content of cytokinins, vitamins, sugars, and other bioactive compounds [28,29]. On the other hand, the spectral quality of light directly influences photosynthesis and microalgae growth. In previous studies, red and blue light are particularly effective in increasing biomass in various species [30]. Finally, $CO_2$ injection was considered as a strategy to increase the availability of inorganic carbon, which can improve the photosynthetic rate and promote faster growth [31].

One of the most relevant findings of this study is the effectiveness of widely available, low-cost, natural substances, such as *Aloe vera* and coconut water, in significantly improving the growth of the microalgae tested. Regarding nutritional composition (oxidizable carbon and nitrogen content), the results showed minimal variations between treatments, with no significant changes compared to the control. Both are raw materials that are accessible in various regions of the world,

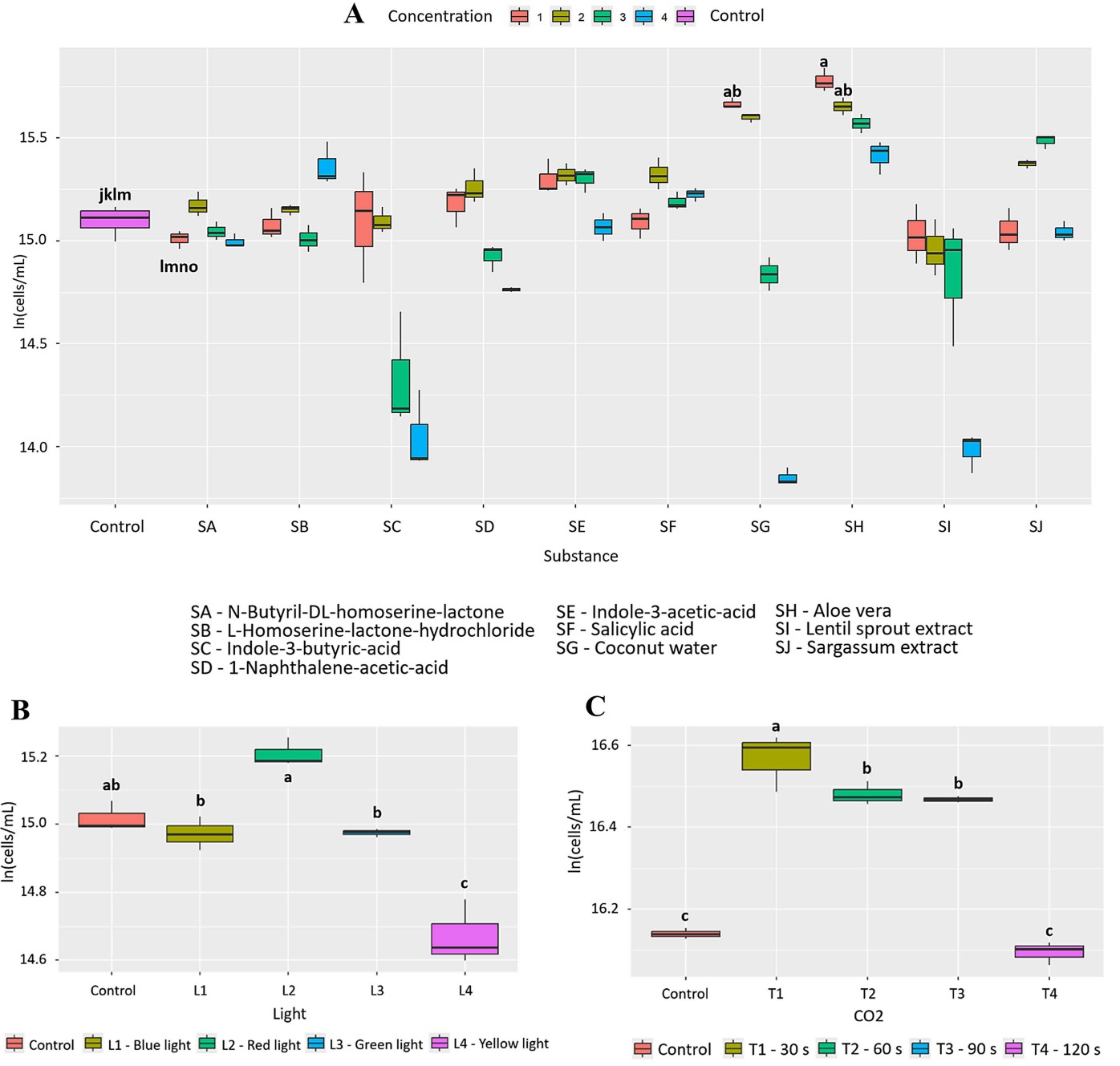

**Fig 6. Effect of the factors evaluated on the growth of *A. falcatus*.** (A) 10 bioactive substances evaluated at 4 concentrations (B) 4 wavelengths and (C) 4 $CO_2$ injection times. Letters indicate statistically significant differences between treatments ($p < 0.05$).

especially in tropical contexts, which facilitates their integration into cultivation systems without requiring expensive technologies or inputs that are difficult to obtain. This accessibility, coupled with their biodegradable nature and traditional use in food and agricultural applications reported in the literature, positions these biostimulants as sustainable alternatives to synthetic or more specialized compounds [32,33]. Together, these results not only reinforce the development of industrial

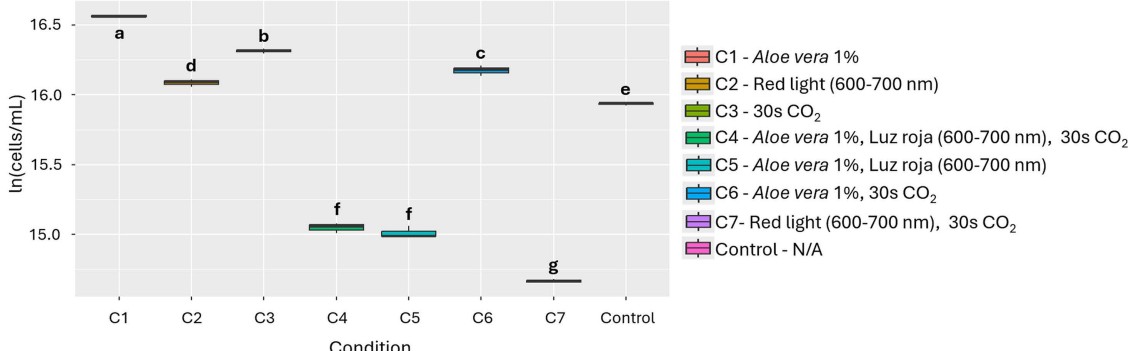

**Fig 7. Effect of the individual and combined use of the 3 evaluated factors on the growth of *A. falcatus*.** Different letters indicate statistically significant differences among treatments (p < 0.05).

solutions with low environmental impact and economic viability, especially in sectors such as aquaculture, agriculture, and water treatment. Together, these results not only reinforce the development of industrial solutions with low environmental impact and economic viability, especially in sectors such as aquaculture, agriculture, and water treatment.

The addition of *Aloe vera* proved to be an effective strategy for stimulating growth, with particularly notable effects on *A. platensis* (PGI = 88.9%), *C. vulgaris* (PGI = 75.0%), and *A. falcatus* (PGI = 85.2%) (Figs 3, 5, 7 and Figs A, B and C in S2 File). This effect can be explained by the presence of bioactive compounds in the extract, such as polysaccharides, vitamins, antioxidants, and minerals, which have been associated with promoting cell proliferation and improving photosynthetic efficiency [34–36]. In addition to growth, changes in nutritional composition were observed (Table 2), such as increases in calcium, phosphorus, iron and zinc in *A. platensis*, total oxidizable organic carbon and potassium in *A. falcatus* and *C. vulgaris*. However, decreases in certain micronutrients, such as calcium, phosphorus, iron, and zinc in *C. vulgaris*, were also recorded, which could indicate internal nutrient redistribution processes or possible competition in their absorption under specific cultivation conditions [22].

Coconut water, on the other hand, showed a notable effect on the growth of *T. dimorphus*, especially at 3%, with a PGI of 80.5% (Fig 9 and Fig D in S2 File), surpassing *Aloe vera* in this species (PGI = 39.8%). This behavior could be due to its rich composition of cytokinins and essential nutrients such as potassium, calcium, and phosphorus, which stimulate key physiological processes such as cell division and photosynthesis [37]. Nutritional improvements were also observed, with increases in calcium, phosphorus, iron, magnesium, potassium, and zinc. However, in other species such as *C. vulgaris* and *A. falcatus*, *Aloe vera* was more efficient, highlighting the need to select biostimulants based on the target species. Other bioactive substances, such as N-Butyryl-DL-homoserine lactone in *A. platensis*, L-Homoserine-lactone hydrochloride in *C. vulgaris*, and plant growth regulators such as naphthaleneacetic acid and indole-3-butyric acid in *T. dimorphus*, showed high efficacy as microalgae growth inducers (Figs 2A, 4A, 6A and Table B in S3 File).

Although the main effect observed was on growth stimulation, the nutritional changes detected were relatively modest and not always consistent across species. This outcome is not unexpected, as enhanced biomass production does not necessarily imply major shifts in cellular composition. In many cases, rapid cell proliferation primarily increases the total amount of biomass available, while the nutrient content per cell remains stable [38,39]. From a biotechnological perspective, this result is still highly relevant, since higher biomass yields are often the primary objective in large-scale microalgae cultivation, and even slight improvements in mineral composition can add value to the final product. Therefore, the findings of this study confirm that while biostimulants such as *Aloe vera* and coconut water can enhance growth efficiency, their nutritional effects may be more subtle and species-dependent.

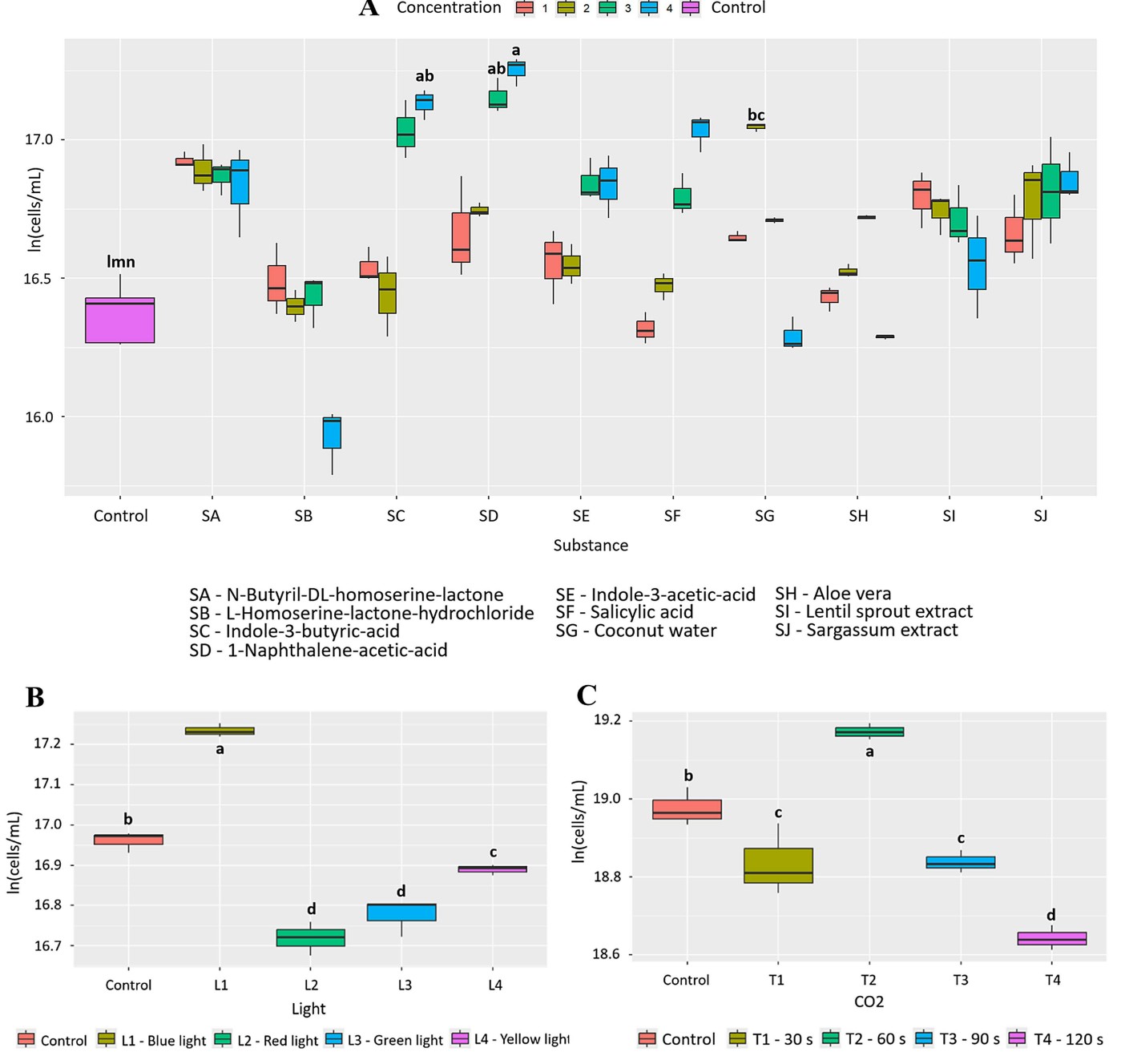

**Fig 8. Effect of the factors evaluated on the growth of *T. dimorphus*.** (A) 10 bioactive substances evaluated at 4 concentrations (B) 4 wavelengths and (C) 4 $CO_2$ injection times. Letters indicate statistically significant differences between treatments ($p < 0.05$).

On the other hand, spectral lighting had a significant and differentiating effect on microalgae growth, highlighting the need to adjust light according to the pigment physiology of each species. Red light (600–700 nm) was most effective in *A. platensis* and *A. falcatus*, with increases of 49.2% and 20.8%, respectively (Figs 2B, 6B, and Table C in S3 File), possibly due to their content of phycobiliproteins such as phycocyanin, which absorb efficiently in that range [40,41]. In contrast, *C.*

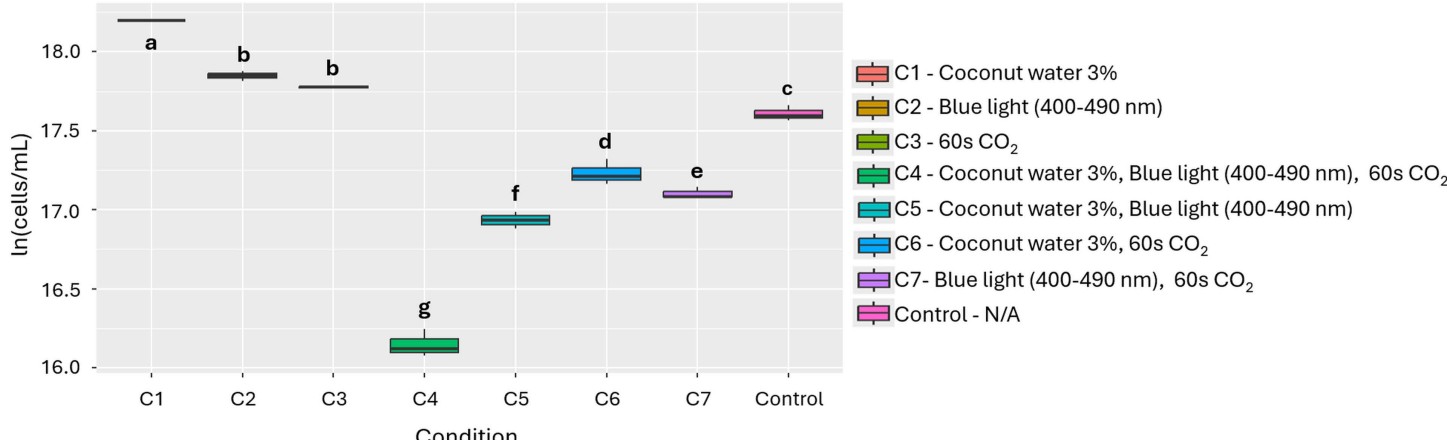

**Fig 9. Effect of the individual and combined use of the 3 evaluated factors on the growth of *T. dimorphus*.** Different letters indicate statistically significant differences among treatments ($p < 0.05$).

*vulgaris* and *T. dimorphus* responded better to blue light (400–490 nm), with increases of 57.7% and 31.5% (Figs 4B, 8B and Table C in S3 File), attributable to their higher proportion of chlorophylls a and b, optimized to capture this spectrum [42,43]. This adaptation could be reinforced by mechanisms such as the xanthophyll cycle or lower sensitivity to photoinhibition, which favors their productivity under light intensity [44].

CO$_2$ supply proved to be an effective stimulus for growth and carbon fixation in the microalgae evaluated, although with different responses depending on the species. *A. platensis, C. vulgaris,* and *T. dimorphus* responded favorably to a 60-second injection (Figs 2C, 4C, 8C and Table D in S3 File), suggesting that intermediate exposures optimize photosynthesis without generating harmful accumulations of dissolved CO$_2$ [45]. On the other hand, *A. falcatus* showed its highest yield with only 30 seconds (Fig 6C and Table D in S3 File), indicating remarkable efficiency in short-term carbon uptake. These differences indicate the need to adjust CO$_2$ injection time according to the species, prioritizing not only biomass yield, but also the efficiency in the assimilation of inorganic carbon [46].

Finally, the combination of stimuli such as bioactive substances, light spectra, and CO$_2$ injection times does not always produce additive or synergistic effects on microalgae growth (Figs 3, 5, 7, 9 and Table E in S3 File). In fact, in species such as *A. platensis* and *A. falcatus*, some combinations (such as condition C6, which includes *Aloe vera* and CO$_2$) showed moderate increases in growth, but were outperformed by individual treatments. Conversely, in species such as *C. vulgaris* and *T. dimorphus*, the combination of materials and conditions often resulted in significant reductions in growth. All this suggests that the simultaneity of stimuli may generate physiological interference or stress effects that inhibit the positive response observed in isolation.

All this indicates that each microalga responds differently and that the interaction between them must be carefully evaluated before scaling up their use in production systems [47,48]. Rather than seeking multiple combinations, it might be more efficient to design specific treatments tailored to the physiological characteristics of each species [49]. Future research should focus on the combined optimization of these materials and conditions, exploring synergies, evaluating their scalability in large-scale production systems, and investigating the underlying biochemical mechanisms to develop more efficient and sustainable strategies for microalgae cultivation.

From a biotechnological perspective, *T. dimorphus* and *A. falcatus* emerge as the most promising species, not only due to their greater responsiveness to individual stimuli, but also due to their nutritional profiles rich in minerals and organic carbon. *T. dimorphus*, in particular, combines accelerated growth with a favorable nutritional profile, while *A. falcatus*

stands out for its carbon fixation efficiency under optimal conditions. These characteristics make them attractive candidates for food, nutraceutical, and environmental mitigation applications.

## 5. Conclusions

Analysis of the four microalgae species evaluated, *Arthrospira platensis, Chlorella vulgaris, Ankistrodesmus falcatus,* and *Tetradesmus dimorphus*, showed significant improvements, primarily in growth under optimal conditions, with no significant changes in nutritional composition compared to the control. The implementation of biostimulants, particularly natural extracts such as *Aloe vera* and coconut water, specific wavelengths, and $CO_2$ injection strategies, proved effective in enhancing microalgae growth, although the response was not always positive when several factors were applied simultaneously. These results highlight the value of integrating biotechnological tools with a detailed understanding of species-dependent responses, where increased biomass is emerging as an initial step for future research aimed at industrial and environmental applications.

## Supporting information

**S1 File. Supplementary figures of cell morphology.**
(DOCX)

**S2 File. Supplementary figures of 30 day growth curves.**
(DOCX)

**S3 File. Supplementary data tables.**
(DOCX)

## Acknowledgments

We extend our gratitude to Professor Fernando Echeverri for his support during the research and data analysis.

## Author contributions

**Formal analysis:** Andrea Llanes.

**Investigation:** Natalia Herrera, Andrea Llanes, Wiston Quiñones.

**Methodology:** Natalia Herrera, Andrea Llanes.

**Project administration:** Natalia Herrera, Wiston Quiñones.

**Supervision:** Natalia Herrera.

**Writing – original draft:** Andrea Llanes.

**Writing – review & editing:** Natalia Herrera, Andrea Llanes.

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
