## [Decision Letter · Decision Letter 0]

4 Feb 2026

Dear Dr. Herrera,

Thank you for submitting your manuscript to PLOS ONE. After careful consideration, we feel that it has merit but does not fully meet PLOS ONE’s publication criteria as it currently stands. Therefore, we invite you to submit a revised version of the manuscript that addresses the points raised during the review process.

We look forward to receiving your revised manuscript.

Kind regards,

Ashfaq Ahmad, Ph.D

Academic Editor

PLOS One

Journal Requirements:

3. Please note that PLOS One has specific guidelines on code sharing for submissions in which author-generated code underpins the findings in the manuscript. In these cases, we expect all author-generated code to be made available without restrictions upon publication of the work. Please review our guidelines at https://journals.plos.org/plosone/s/materials-and-software-sharing#loc-sharing-code and ensure that your code is shared in a way that follows best practice and facilitates reproducibility and reuse.

5. We note that Figure 1 in your submission contain copyrighted images. All PLOS content is published under the Creative Commons Attribution License (CC BY 4.0), which means that the manuscript, images, and Supporting Information files will be freely available online, and any third party is permitted to access, download, copy, distribute, and use these materials in any way, even commercially, with proper attribution. For more information, see our copyright guidelines: http://journals.plos.org/plosone/s/licenses-and-copyright.

6. Please ensure that you refer to Figure 9 in your text as, if accepted, production will need this reference to link the reader to the figure.

Reviewers' comments:

Reviewer's Responses to Questions

**Comments to the Author**

1. Is the manuscript technically sound, and do the data support the conclusions?

Reviewer #1: Yes

Reviewer #2: Yes

Reviewer #3: Yes

2. Has the statistical analysis been performed appropriately and rigorously?

Reviewer #1: I Don't Know

Reviewer #2: Yes

Reviewer #3: Yes

3. Have the authors made all data underlying the findings in their manuscript fully available?

Reviewer #1: Yes

Reviewer #2: Yes

Reviewer #3: Yes

4. Is the manuscript presented in an intelligible fashion and written in standard English?

Reviewer #1: Yes

Reviewer #2: Yes

Reviewer #3: Yes

Reviewer #1: General Comments

The manuscript presents a comprehensive experimental study systematically evaluating the effects of bioactive substances, light wavelength, and CO₂ injection regimes—both individually and in combination—on the growth of four microalgae/cyanobacterium of biotechnological interest (Arthrospira platensis, Chlorella vulgaris, Ankistrodesmus falcatus, and Tetradesmus dimorphus).

The inclusion of low-cost alternatives (such as Aloe vera and coconut water) is relevant from an applied perspective, and all data are fully available in the supplementary files, in accordance with the journal’s data availability requirements.

Despite its experimental merit, the manuscript presents several important issues that need to be addressed before it can be considered for publication, including:

Textual clarity and English quality:

There are recurrent grammatical errors, inappropriate use of prepositions, agreement issues, and long or confusing sentence constructions. In several sections, the English does not fully compromise comprehension, but it does reduce scientific clarity, particularly in the Introduction and Discussion.

Conceptual coherence between objectives, methods, and conclusions:

The stated objective refers to growth “and/or nutritional quality,” but the nutritional analysis is limited and applied only to selected conditions. While the conclusions emphasize growth, the nutritional data show only modest changes.

Statistical interpretation and risk of inflated significance:

The extensive use of the LSD test in an experimental design involving many treatments increases the risk of Type I error. Although statistical assumptions were validated, the choice of the LSD test should be more clearly justified given the large number of comparisons.

The specific points that require clarification and revision are detailed below.

Abstract

1. The abstract presents a large number of percentage values, making it dense and difficult to read. Reducing the number of quantitative examples and retaining only the most representative ones is recommended.

2. The terms “CO₂ injection regimes,” “CO₂ injection times,” and “CO₂ injection” are used interchangeably. Terminology should be standardized throughout the manuscript.

3. The abstract suggests the investigation of synergistic effects; however, the results clearly indicate a lack of additive effects. This nuance should be stated more explicitly and with a less optimistic tone.

Introduction

1. There is repetition of ideas. The relevance of microalgae and growth optimization is reiterated across multiple paragraphs with limited conceptual progression. For example, lines 37–45 and 47–49 essentially address the same argument.

2. The final objective of the study (lines 65–70) refers to both growth and nutritional quality, which creates expectations of more in-depth nutritional analyses than those performed. The objective should be clarified, emphasizing that growth is the main focus and that nutritional analysis is exploratory or complementary.

Materials and Methods

1. This section contains excessive operational detail.

2. The description of culture isolation and cleaning procedures is too long for a research article.

3. It is recommended to condense operational details, retaining only what is essential for reproducibility.

4. It should be clearly stated that species identification was based on morphology only, without molecular confirmation.

5. The rationale for selecting the concentration ranges of bioactive substances is not clearly explained.

6. Some substances show positive effects only at very specific concentrations, but this is not discussed methodologically.

7. References should be included to justify the selected concentrations, or the authors should explicitly state that these were exploratory concentrations.

8. Terminology should be standardized, for example:

“Indol-3-butíric acid” vs. “Indole-3-butyric acid”;

“naphtalenacético” vs. “naphthaleneacetic”.

9. Light intensity is described as “estimated” (80–150 µmol m⁻² s⁻¹). Is there a limitation preventing direct measurement of this variable? If intensity was not controlled, conclusions based solely on spectral effects should be interpreted cautiously.

10. Why is biomass recovery from the culture medium described only in Section 2.5?

11. Was the same wavelength (630 nm) used for biomass determination for all species? This wavelength may not be optimal for all microalgae analyzed.

12. Is there a bibliographic reference supporting the method used to construct the calibration curves? Were the calibration curves established after 15 days of growth monitoring?

13. Why were only one initial point and one final point (12 days) used for the main tests?

14. Regarding statistical analysis, why was a more conservative post hoc test (e.g., Tukey) not applied?

Results

1. The resolution of Figures 2–9 should be improved.

2. Many figures contain excessive visual information (numerous LSD test letters), making interpretation tiring. Simplification of the graphical presentation is recommended, if possible.

3. Growth curves are not fully explored in the text, representing a missed opportunity to enrich the dynamic interpretation of growth.

4. There is excessive repetition of numerical values, p-values, and LSD letters in the text. Reducing numerical detail and emphasizing general trends is suggested.

5. The Results section contains limited interpretative synthesis, with extensive description and little integration across findings. Better integration of results is recommended.

6. Information presented in figures and tables is extensively repeated in the text. Reorganization is suggested to avoid redundancy.

7. It should be clearly stated that combined treatments are not necessarily advantageous and may induce physiological stress.

8. Why is no statistical analysis presented for the responses shown in Table 2?

9. The term “significant nutritional improvement” should be avoided unless supported by appropriate statistical analysis, which should be included if available.

Discussion

1. The Discussion includes repeated ideas, long paragraphs, and occasionally redundant language, which compromises fluency and objectivity. Revision is recommended to remove repetition and synthesize arguments.

2. In some passages, results are presented as robust evidence of physiological mechanisms, whereas the data are essentially observational. Such statements should be revised. For example, direct attribution of effects to cytokinins, vitamins, or antioxidants is made without direct biochemical measurements. More cautious language is recommended (e.g., “may be associated with,” “possibly related to”).

3. The relevance of Aloe vera and coconut water as sustainable alternatives is reiterated several times using similar arguments.

4. Light intensity was not rigorously controlled, only estimated, and CO₂ injection was based on time rather than dissolved concentration. These limitations should be more explicitly acknowledged to avoid overinterpretation of the results.

5. The discussion of combined factors (bioactives + light + CO₂) could be more concise and more directly connected to the data. The central argument (physiological interference and non-linear responses) becomes diluted in lengthy paragraphs.

Conclusions

1. The conclusions refer to “nutritional quality,” but the data show modest and inconsistent changes. Reducing the emphasis on this aspect or explicitly characterizing it as exploratory is recommended.

2. The conclusions could more strongly emphasize that the results are specific to the experimental conditions adopted in this study.

Reviewer #2: The manuscript entitled “Enhancing microalgal productivity through bioactive substances, light, and CO₂”. This manuscript need revision before publication

Comments

1. Line 86: and remove unwanted particles. What are the unwanted martials and how authors remove them in microalgae culture?

2. Line 86-90 are confusing need to improve it.

3. Lin 93 what was the concentration of microalgae (OD) in the start of the experiment.

4. Why authors use such low light intensity (100 μmol m⁻² s⁻¹,) for low 12 h:12 h light/dark photoperiod. The optimized is 18:6.

5. How authors determine the growth of microalgae.

6. Table 1..why authors choose such costly bioactive compound rather than any other economic material for imrove the growth of microalgae. Is this study is economic feasible.

7. Where is the growth curve. Provide it in supplementary material.

8. Authors may add the flowing publication in discussion to improve the quality of paper

10.1016/j.pbi.2025.102696, 10.1016/j.bcab.2024.103315, 10.3390/su16167075,

9. Figures are not in good resolution. Provide the figure with good resolution.

Reviewer #3: This article addresses an interesting and important topic: the search for conditions that enhance the productivity of microalgae, which in turn are a promising resource for the production of valuable products, biofuels, and other materials. The article presents a methodology and applies an interesting approach that allows to determine the influence of individual growth factors (stimuli) and to study the complex (simultaneous) impact of several factors.

The following comments need to be made:

1. The authors should note that the introduction, specifically the analysis of previously conducted studies, is extremely brief. It is important to present not only the general results of such studies but also a brief description of the methods used. It would be desirable to include in the introduction the results of the combined effects of various growth factors on microalgal biomass, if such studies have been conducted previously.

2. Given the low solubility of CO2, it is unclear what processes could have achieved any noticeable changes in microalgal biomass productivity with such short exposure times (30 to 120 s). Experiments assessing the effect of CO2 on microalgal growth are typically conducted with continuous bubbling of the culture fluid with a mixture of CO2 and air with an elevated carbon dioxide concentration. Can the authors explain why this particular mode of CO2 addition to the culture fluid was chosen for the experiments?

3. The authors should clarify the composition of the gases during aeration during the main part of the experiments – was it air with a standard CO2 concentration? In section 2.2, the authors write: “Aeration was continuous and maintained at 1.0 L/min.”

4. Figures labeled "A" are too information-heavy and difficult to understand and analyze. Perhaps a more detailed analysis should be provided in Table B in File S3?

.

Reviewer #1: No

Reviewer #2: No

Reviewer #3: **Yes:**Sofia KiselevaSofia KiselevaSofia KiselevaSofia Kiseleva

---

## [Author Response · Author response to Decision Letter 1]

5 Mar 2026

Editorial requirement:

Figure 1 has been revised. The figure corresponds to an original photograph taken by the authors during their own experimental assays; therefore, no third-party copyrighted material is included.

Reviewer #1:

The manuscript presents a comprehensive experimental study systematically evaluating the effects of bioactive substances, light wavelength, and CO₂ injection regimes—both individually and in combination—on the growth of four microalgae/cyanobacterium of biotechnological interest (Arthrospira platensis, Chlorella vulgaris, Ankistrodesmus falcatus, and Tetradesmus dimorphus).

The inclusion of low-cost alternatives (such as Aloe vera and coconut water) is relevant from an applied perspective, and all data are fully available in the supplementary files, in accordance with the journal’s data availability requirements. Despite its experimental merit, the manuscript presents several important issues that need to be addressed before it can be considered for publication, including: Textual clarity and English quality: There are recurrent grammatical errors, inappropriate use of prepositions, agreement issues, and long or confusing sentence constructions. In several sections, the English does not fully compromise comprehension, but it does reduce scientific clarity, particularly in the Introduction and Discussion. Conceptual coherence between objectives, methods, and conclusions: The stated objective refers to growth “and/or nutritional quality,” but the nutritional analysis is limited and applied only to selected conditions. While the conclusions emphasize growth, the nutritional data show only modest changes. Statistical interpretation and risk of inflated significance: The extensive use of the LSD test in an experimental design involving many treatments increases the risk of Type I error. Although statistical assumptions were validated, the choice of the LSD test should be more clearly justified given the large number of comparisons. The specific points that require clarification and revision are detailed below.

Abstract

1. The abstract presents a large number of percentage values, making it dense and difficult to read. Reducing the number of quantitative examples and retaining only the most representative ones is recommended.

The abstract has been revised to substantially reduce the number of quantitative values, retaining only representative trends and emphasizing general patterns rather than detailed percentages, in order to improve clarity and readability.

Lines 12-35.

2. The terms “CO₂ injection regimes,” “CO₂ injection times,” and “CO₂ injection” are used interchangeably. Terminology should be standardized throughout the manuscript.

Terminology has been standardized throughout the abstract and manuscript, and the term “CO₂ injection times” is now used consistently.

Line 28.

3. The abstract suggests the investigation of synergistic effects; however, the results clearly indicate a lack of additive effects. This nuance should be stated more explicitly and with a less optimistic tone.

The abstract has been revised to explicitly state that combined treatments did not produce additive or synergistic effects, and the tone has been adjusted to reflect a more cautious interpretation of the results.

Line 30.

Introduction

1. There is repetition of ideas. The relevance of microalgae and growth optimization is reiterated across multiple paragraphs with limited conceptual progression. For example, lines 37–45 and 47–49 essentially address the same argument.

We have revised the Introduction to eliminate repetitive statements regarding the relevance of microalgae and biomass optimization. The section has been reorganized to improve conceptual progression, moving from the ecological and biotechnological importance of microalgae to the necessity of optimizing growth for scalable applications, followed by current knowledge gaps.

Lines 42-51.

2. The final objective of the study (lines 65–70) refers to both growth and nutritional quality, which creates expectations of more in-depth nutritional analyses than those performed. The objective should be clarified, emphasizing that growth is the main focus and that nutritional analysis is exploratory or complementary.

The study objective has been revised to emphasize that growth performance and biomass enhancement constitute the primary focus of the work. Nutritional parameters are now explicitly described as complementary indicators used to contextualize growth responses, rather than as a central analytical objective. This adjustment better aligns the stated objective with the scope of the results presented.

Lines 95-102.

Materials and Methods

1. This section contains excessive operational detail.

The Materials and Methods section has been revised to reduce excessive operational detail. Non-essential procedural descriptions have been condensed while retaining all information necessary for reproducibility.

2. The description of culture isolation and cleaning procedures is too long for a research article.

The description of culture isolation and cleaning procedures has been substantially condensed to include only key steps required for understanding and reproducibility.

Lines 113-119.

3. It is recommended to condense operational details, retaining only what is essential for reproducibility.

Operational descriptions have been streamlined throughout the section. Only essential experimental conditions, culture parameters, and analytical procedures have been retained.

4. It should be clearly stated that species identification was based on morphology only, without molecular confirmation.

The manuscript has been revised to explicitly state that species identification was based exclusively on morphological characteristics following taxonomic keys and Algaebase references, without molecular confirmation.

Lines 130-133.

5. The rationale for selecting the concentration ranges of bioactive substances is not clearly explained.

A clarification has been added to section 2.3 explaining that the selected concentration ranges were defined based on preliminary exploratory assays and literature reports describing biological activity at low concentrations in microalgal systems.

Lines 186-196.

6. Some substances show positive effects only at very specific concentrations, but this is not discussed methodologically.

The methodology has been revised to clarify that the selected concentration ranges were intended to identify potential stimulatory windows, acknowledging that positive effects may occur only at specific concentrations.

Lines 186-196.

7. References should be included to justify the selected concentrations, or the authors should explicitly state that these were exploratory concentrations.

The manuscript now specifies that the tested concentrations were exploratory and informed by literature describing biological activity ranges of similar compounds in microalgal systems.

Lines 186-196.

8. Terminology should be standardized, for example: “Indol-3-butíric acid” vs. “Indole-3-butyric acid”; “naphtalenacético” vs. “naphthaleneacetic”.

Terminology has been standardized throughout the manuscript. All compound names have been revised to ensure consistent English nomenclature.

Line 198 (Table 1).

9. Light intensity is described as “estimated” (80–150 μmol m⁻² s⁻¹). Is there a limitation preventing direct measurement of this variable? If intensity was not controlled, conclusions based solely on spectral effects should be interpreted cautiously.

In the revised manuscript, light intensity was directly measured at the culture surface using a lux meter. The recorded values were approximately 170 lux (red), 144 lux (blue), 273 lux (green), 280 lux (yellow), and 283 lux (white control). We acknowledge that lux measurements do not directly correspond to photosynthetically active radiation (PAR); however, identical LED configurations and reflective chamber geometry were maintained across treatments to ensure comparable illumination conditions. The manuscript has been updated in Section 2.4 to report the measured values and clarify the basis for spectral comparisons. Interpretations remain focused primarily on spectral distribution under controlled experimental settings.

Lines 217-224.

Lines 669-671.

10. Why is biomass recovery from the culture medium described only in Section 2.5?

We appreciate the reviewer’s observation. Biomass recovery was described in Section 2.5 because dry weight determination was required specifically for CO₂ fixation calculations. Routine growth assessment throughout the study was performed using cell density measurements obtained by spectrophotometry and cell counting (Section 2.6), and therefore biomass harvesting was not necessary for those analyses. To improve clarity, we have added a statement in Section 2.6 explaining that dry biomass determination was conducted exclusively for CO₂ fixation analysis.

Lines 245-248.

Lines 295-297.

11. Was the same wavelength (630 nm) used for biomass determination for all species? This wavelength may not be optimal for all microalgae analyzed.

Optical density at 630 nm was selected because it is commonly used for estimating microalgal biomass and provides reliable correlations with cell density while minimizing interference from chlorophyll absorption peaks. Importantly, species-specific calibration curves were constructed for each microalga by correlating OD₆₃₀ with direct cell counts. This approach ensured accurate biomass estimation for each species independently, regardless of potential differences in spectral absorption characteristics. We have clarified this justification in Section 2.6 and included additional supporting references.

Lines 252-255.

12. Is there a bibliographic reference supporting the method used to construct the calibration curves? Were the calibration curves established after 15 days of growth monitoring?

The calibration curves were constructed by correlating OD₆₃₀ measurements with direct cell counts over a 30-day monitoring period, covering different growth phases. Measurements were performed every 24 h during the first 15 days to accurately characterize the exponential phase, and every 48 h thereafter. The calibration equations were derived from linear regression analyses between optical density and cell density. This approach is consistent with previously reported spectrophotometric biomass estimation methods in microalgae. We have clarified this description in Section 2.6 and included additional supporting references.

Lines 252-259 and 266-271.

13. Why were only one initial point and one final point (12 days) used for the main tests?

The 12-day experimental period was selected based on prior exploratory assays conducted under similar culture conditions and is consistent with previous studies from our research group evaluating microalgal growth responses over 10-day periods under comparable setups [1,2]. These studies indicate that treatment effects are typically most evident during the late exponential phase (approximately days 10–15). Additionally, extended 30-day assays (included in the Supplementary Material) confirmed that maximal differences among treatments occurred within this time window, while later measurements reflected stationary-phase stabilization. We have clarified this rationale in the revised manuscript.

1. Herrera N, Echeverri F. Evidence of Quorum Sensing in Cyanobacteria by Homoserine Lactones: The Origin of Blooms. Water. 2021; 13:1831. https://doi.org/10.3390/w13131831

2. Herrera N, Llanes A, Echeverri F. Effect of pesticides and tributaries on the growth and development of Microcystis aeruginosa and Cylindrospermopsis raciborskii. Algal Research. 2025; 85:103876. https://doi.org/10.1016/j.algal.2024.103876

Lines 153-160.

14. Regarding statistical analysis, why was a more conservative post hoc test (e.g., Tukey) not applied?

Fisher’s LSD test was applied only after verification of model assumptions and a significant ANOVA for each independent linear model (bioactive substances, wavelengths, CO₂ treatments, and combined conditions) and species. Although the bioactive substances factor involved multiple treatments, analyses were conducted within each species and factor separately using a balanced design with triplicates. LSD was selected to maintain statistical power for detecting biologically meaningful differences relative to the control. Furthermore, the magnitude of the observed effects and the highly significant p-values (S3 File. Supplementary data tables) indicate robust treatment differences. The statistical analysis section has been clarified accordingly.

Lines 357-375.

Results

1. The resolution of Figures 2–9 should be improved.

All figures have been replaced with high-resolution versions to improve clarity and readability.

2. Many figures contain excessive visual information (numerous LSD test letters), making interpretation tiring. Simplification of the graphical presentation is recommended, if possible.

The graphical presentation has been simplified to improve readability. The number of statistical letters displayed has been reduced where possible, and emphasis has been placed on the most relevant treatment comparisons.

3. Growth curves are not fully explored in the text, representing a missed opportunity to enrich the dynamic interpretation of growth.

A brief description of the growth dynamics has been incorporated into the Results section to highlight differences in lag phase duration, exponential growth behavior, and stabilization patterns observed among treatments.

Lines 396-399.

Lines 438-440.

Lines 467-469.

Lines 504-505.

4. There is excessive repetition of numerical values, p-values, and LSD letters in the text. Reducing numerical detail and emphasizing general trends is suggested.

The Results section has been revised to reduce excessive numerical repetition.

5. The Results section contains limited interpretative synthesis, with extensive description and little integration across findings. Better integration of results is recommended.

The Results section has been revised to improve integration across species and experimental factors. Comparative trends (e.g., common positive effects of Aloe vera or blue/red light across species) are now briefly highlighted to provide a more cohesive presentation of the findings while maintaining separation from the Discussion section.

6. Information presented in figures and tables is extensively repeated in the text. Reorganization is suggested to avoid redundancy.

We agree and have reorganized the Results section to avoid redundancy.

7. It should be clearly stated that combined treatments are not necessarily advantageous and may induce physiological stress.

The Results section now clearly states that combined treatments were not consistently advantageous and, in certain cases, resulted in lower growth responses compared to individual treatments, suggesting potential physiological stress effects.

Lines 406-409.

Lines 442-444.

Lines 480-482.

Lines 517-519.

8. Why is no statistical analysis presented for the responses shown in Table 2?

The compositional analysis presented in Table 2 was conducted on biomass obtained under the best-performing condition for each species. Due to the nature of the analytical procedure, measurements represent single determinations per condition and are therefore presented as descriptive data. This has been clarified in the revised manuscript.

Lines 539-544.

9. The term “significant nutritional improvement” should be avoided unless supported by appropriate statistical analysis, which should be included if available.

The term “nutritional improvement” has been removed and replaced with descriptive language.

Line 765-766.

Discussion

1. The Discussion includes repeated ideas, long paragraphs, and occasionally redundant language, which compromises fluency and objectivity. Revision is recommended to remove repetition and synthesize arguments.

The Discussion section has been revised to improve clarity and conciseness. Repeated ideas have been removed, lengthy paragraphs have been divid

---

## [Decision Letter · Decision Letter 1]

25 Mar 2026

Enhancing microalgal productivity through bioactive substances, light, and CO₂

PONE-D-25-63084R1

Dear Dr. Herrera,

We’re pleased to inform you that your manuscript has been judged scientifically suitable for publication and will be formally accepted for publication once it meets all outstanding technical requirements.

Kind regards,

Ashfaq Ahmad, Ph.D

Academic Editor

PLOS One

Additional Editor Comments (optional):

Reviewers' comments:

Reviewer's Responses to Questions

**Comments to the Author**

Reviewer #2: All comments have been addressed

Reviewer #3: All comments have been addressed

2. Is the manuscript technically sound, and do the data support the conclusions?

Reviewer #2: Yes

Reviewer #3: Yes

3. Has the statistical analysis been performed appropriately and rigorously?

Reviewer #2: Yes

Reviewer #3: Yes

4. Have the authors made all data underlying the findings in their manuscript fully available?

Reviewer #2: Yes

Reviewer #3: Yes

5. Is the manuscript presented in an intelligible fashion and written in standard English?

Reviewer #2: Yes

Reviewer #3: Yes

Reviewer #2: In the revised submission the authors have satisfactorily addressed my comments and concerns raised on their original submission. Therefore, I recommend publication.

Reviewer #3: The authors provided adequate responses to all the reviewer's comments. I disagree with some of the author's judgments (comment No. 2), but I admit that this experimental design (short-term bubbling of a microalgae suspension with 100% carbon dioxide) could be considered in a scientific study.

The description of the experimental methodology is adequate and complete. The experiments were conducted in at least three replicates. Statistical processing of the results was performed. The results are presented in detail, analyzed, and discussed.

.

Reviewer #2: No

Reviewer #3: No

---

## [Editor Report · Acceptance letter]

PONE-D-25-63084R1

PLOS One

Dear Dr. Herrera,

I'm pleased to inform you that your manuscript has been deemed suitable for publication in PLOS One. Congratulations! Your manuscript is now being handed over to our production team.

Kind regards,

on behalf of

Dr. Ashfaq Ahmad

Academic Editor

PLOS One